# A landscape of complex tandem repeats within individual human genomes

Kazuki Ichikawa[1], Riki Kawahara[1], Takeshi Asano[1] & Shinichi Morishita [1] ✉

Markedly expanded tandem repeats (TRs) have been correlated with ~60 diseases. TR diversity has been considered a clue toward understanding missing heritability. However, haplotype-resolved long TRs remain mostly hidden or blacked out because their complex structures (TRs composed of various units and minisatellites containing >10-bp units) make them difficult to determine accurately with existing methods. Here, using a high-precision algorithm to determine complex TR structures from long, accurate reads of PacBio HiFi, an investigation of 270 Japanese control samples yields several genome-wide findings. Approximately 322,000 TRs are difficult to impute from the surrounding single-nucleotide variants. Greater genetic divergence of TR loci is significantly correlated with more events of younger replication slippage. Complex TRs are more abundant than single-unit TRs, and a tendency for complex TRs to consist of <10-bp units and single-unit TRs to be minisatellites is statistically significant at loci with ≥500-bp TRs. Of note, 8909 loci with extended TRs (>100b longer than the mode) contain several known disease-associated TRs and are considered candidates for association with disorders. Overall, complex TRs and minisatellites are found to be abundant and diverse, even in genetically small Japanese populations, yielding insights into the landscape of long TRs.

Tandem repeats (TRs) are genomic sequences in which one or more string units are present in tandem[1]. During the past 30 years, over sixty diseases have been associated with markedly expanded TRs at different loci[2] and TR diversity has been considered a clue toward understanding missing heritability[2]. TRs consisting of short units (2–6 bp) were discovered in the early 1980s; they were initially called microsatellites[3–5], and later termed short sequence repeats (SSRs), short tandem repeats (STRs), or simple repeats[6]. In 1985, TRs with longer units of a few dozen bases were also discovered and were called minisatellites[7]. Micro- and minisatellites that vary in length among individuals and are prone to variants are referred to as variable number tandem repeats (VNTRs)[8], which are valuable for investigating genetic diversity in human populations[9,10].

TR stretching is thought to be driven by replication slippage or non-homologous recombination[8], but replication slippage can also lead to the shortening of long TRs[11]. TR expansion is not repeated forever but can be halted by the presence of point mutations[12]. The inherent nature of such TRs was observed in disease-specific regions through family-driven approaches when genome-wide sequence data were difficult to collect. When abundant short-read genomic data such as individual exome and whole genome sequences became available for populations, algorithms were proposed to estimate the lengths and structures of TRs in individual genomes on a genome-wide basis[13–17], although it remains difficult to accurately determine the entire structure of TRs of >100-bp in length using short-read sequencing. Therefore, long-read sequencing approaches such as PacBio and Nanopore have become attractive because they can cover most TRs <10 kb in length and can sequence long DNA fragments without using polymerase chain reaction, which is prone to replication slippage during amplification[18]. An initial study using long-read sequencing suggested that ~30% of structural variants are TRs[19].

[1]Department of Computational Biology and Medical Sciences, The University of Tokyo, 277-8561 Chiba, Japan. ✉e-mail: moris@edu.k.u-tokyo.ac.jp

Long-read sequencing is now expected to reveal two types of hidden, disease-associated TRs: minisatellites and complex TRs. A couple of disease-associated minisatellites have been identified recently; for example, 100–3000 30-mer copies were found to be associated with schizophrenia and bipolar disorder[20], over 200 25-mer copies in *ABCA7* were found to be specific to Alzheimer's disease with an odds ratio of 4.5[21], and 69-mer repeats in *WDR7* were identified in amyotrophic lateral sclerosis (ALS)[22]. Copies of long units in minisatellites often contain variants and therefore are difficult to identify using conventional tools such as TRF[23], necessitating the development of tools such as mTR, to accurately identify minisatellites[24].

Another type of disease-associated TR has a complex structure, in which different units are expanded within personal genomes. To date, several disease-associated complex TRs have been reported, including 400–2000 copies of AAAAG, AAAGG, AAGAG, and AGAGC in *RFC1*, which are associated with cerebellar ataxia, neuropathy, vestibular areflexia syndrome (CANVAS)[25]; the loss of CAA and CCA in (CAG)m CAA CAG CCA (CCG)n, the motif structure of *HTT* gene, are associated with Huntington's disease (HD)[26], CAG and ACT in *ATXN8* are with spinocerebellar ataxia type 8 (SCA8)[27], CAGG, CA, and CAGA in *CNBP* with myotonic dystrophy type 2 (DM2)[28], and TTTCA and TTTTA in *SAMD12* with benign adult familial myoclonic epilepsy (BAFME)[29].

It is difficult to automatically determine the sequence compositions of complex TRs, which are typically confirmed by manual inspection. Tools such as PacBio structural variant (PBSV), TRF[23], RepeatMasker (http://repeatmasker.org) have been widely used to detect structural variants but are not designed to determine complex TR structure and are not suited to correctly parse complex TRs. For example, repeats of the form (AAAG)$i$ (AG)$j$ (AGGG)$k$ (AG)$l$ (AAAG)$m$, where suffixes indicate unit occurrences, are likely to be falsely detected as single (AAAG)-repeats. These methodological issues make it difficult to systematically examine the contribution of TRs to human disorders on a genome-wide scale[2].

In this work, using our mathematical models and efficient algorithms[24,30] to address the problem of automatic, accurate determination of complex TRs and minisatellites, we will demonstrate that complex TRs and minisatellites are highly diverse even in genetically small Japanese populations, thereby providing a landscape of complex TRs hidden within individual human genomes.

## Results

### Long-read sequencing of 270 individuals
We conducted HiFi sequencing using a PacBio Sequel II system to collect highly accurate (~99.9% base accuracy) long reads (median length of 14,118 bp) from B cells derived from 270 healthy Japanese subjects (see the length distribution in Supplementary Fig. 1). Note here that specific TR alleles may be selected for during the immortalization process by EBV and do not necessarily represent the original allele. Although separate haplotype sequencing is ideal, diploid genome assembly tools require 30- to 40-fold coverage of reads from a single individual[31]. To obtain data from more individuals, we used one single-molecule real-time sequencing cell (SMRT cell) to obtain a coverage of ≥7.5 reads, theoretically allowing the observation of complex TRs with lengths of 5, 2, and 1 kb within one of two haplotypes with ≥1 read of 14 kb read at probability levels of 91%, 96%, and 97%, respectively, according to the Lander-Waterman statistics ("Methods", Supplementary Fig. 2). Nanopore ultralong sequencing was also considered as a candidate; however, its accuracy (95–99%) limited its ability to distinguish between variants and sequencing errors. Moreover, a tremendous amount of Nanopore reads are needed to correct highly erroneous reads; for example, Mizuguchi et al. collected ~470-fold coverage of Nanopore reads from a region of a single gene that is known to have disease-associated TRs[32].

### Automatic decomposition of complex TRs
For the automatic characterization of complex TRs, we used our algorithm that decomposes genomic regions and reads containing TRs into a series of neighboring substrings (repeat units)[30]. Many such decompositions are possible, and it is crucial to select the optimal decomposition by assessing the goodness of each one. Assuming maximum parsimony that prefers fewer events of replication slippage and/or non-homologous recombination, we defined a penalty that becomes smaller when a decomposition consists of fewer copies of fewer, shorter generating units. We developed a computationally efficient algorithm that outputs a decomposition with nearly minimum penalty. Using synthetic benchmark data, we confirmed the practical feasibility of detecting typical complex TRs in human genomes with almost 100% accuracy and the computational efficiency of processing complex TRs in time linear to length ("Methods")[30].

Figure 1a shows examples of complex TRs containing four different units in intron 2 of *RFC1*, where a biallelic AAGGG repeat expansion is associated with CANVAS[25]. Two groups of complex TRs were categorized by the absence or presence of the single-nucleotide variants (SNV) closest to the TRs, indicated by Groups I and II, respectively. TRs in Group II were identified in our samples using our algorithm. Each group comprises several different subgroups, and their decompositions are illustrated as waves on the right side of the table. The rightmost column shows the number of occurrences of each pattern in all alleles; for example, even one occurrence is guaranteed by multiple independent HiFi reads. Notably, repeats with ACAGG units (red) in Group II are markedly expanded, with 586 copies of the unit, and the subgroups within Group II are indistinguishable from pairs of the nearest-neighbor SNV (black). This example suggests that many different complex TR subtypes can coexist in the same linkage disequilibrium block and that TRs can have greater genetic divergence than the surrounding SNVs. To confirm this tendency, we mapped long reads with TRs from individuals to the reference genome, and examined the sequence compositions of TRs at each locus.

### Listing candidate regions with TRs in the reference human genome (hg38)
We listed candidate loci with TRs (hereafter, TR loci) in the reference human genome (hg38) using two methods, mTR and RepeatMasker, which are complementary in enumerating TRs. RepeatMasker is good at listing microsatellites, whereas mTR can also list minisatellites whose units are longer than 10 bp[24]. Supplementary Fig. 3 shows a breakdown by length of TR loci in the reference genome and classifies each length range into three groups, depending on the TR locus detected by mTR alone, RepeatMasker alone, or both. More than half of the TRs longer than 500 bp were detected by mTR alone, most of which were minisatellites with units longer than 10 bp. Most of the other TRs detected using both methods or by RepeatMasker alone had units shorter than 6 bp. For detecting minisatellites, TRF has been also widely used[23]. Although mTR and estimated TRs of approximately the same length (Supplementary Fig. 3d, "Methods"), mTR was likely to output longer TRs with shorter units (Supplementary Fig. 3d, e), and hence TRs estimated by mTR were used in this study. A total of 2,202,622 TR loci were identified, of which 121,556 (~5.5%) were located in segmental duplications in the hg38 reference (hg38_genomicSuperDups_2014-10-14), so to avoid false read alignments in segmental duplications and were excluded from the analysis in this study.

### Detection of TR alleles from reads
To study the TR distribution at each TR locus, reads from one individual were anchored to candidate TR loci. Reads at each locus were grouped in terms of sequence similarity to observe one or two groups (denoted by TR alleles in Fig. 1b) that respectively represent homozygous or heterozygous for TRs. Supplementary Fig. 4a shows the distribution of the ratio of observed TR alleles to total alleles at all TR

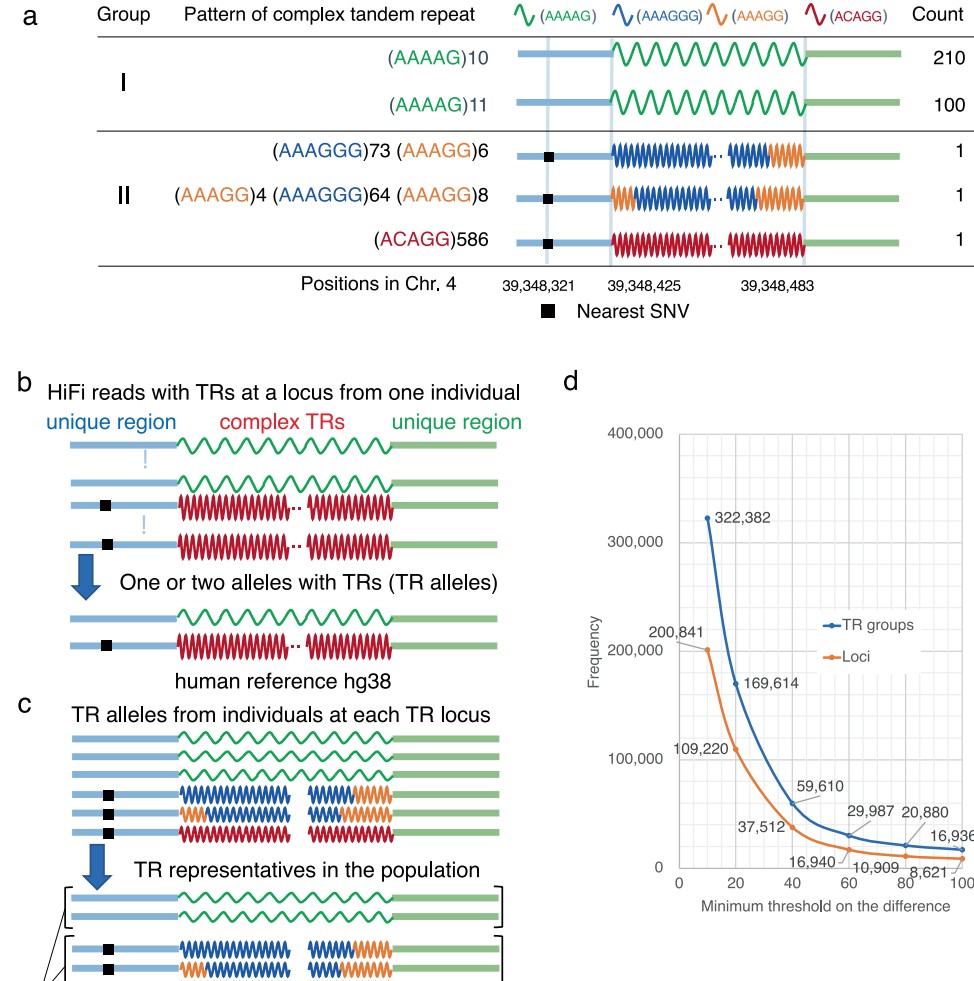

**Fig. 1 | Detection of complex tandem repeats outside segmental duplications in individual genomes. a** Examples of complex tandem repeats in an intron of *RFC1*. The first column shows the two groups of TRs classified by the surrounding nearest-neighbor SNVs shown in the 3rd column (group I is identical to the reference). The 2nd column shows five examples of tandem repeat patterns, and the 3rd and 4th tandem repeats are complex. The 3rd column illustrates each pattern by colored waves associated with the nearest-neighbor SNVs. The last shows the count of each pattern in our study. Each pattern is confirmed by multiple HiFi reads. **b** HiFi reads with TRs from each individual are anchored to the human genome reference (hg38). Loci with TRs are called TR loci. Reads are clustered into one or two alleles with TRs (denoted as TR alleles) at each TR locus according to sequence similarity. The SNVs closest to the TRs are searched in the peripheral region (indicated by black squares). **c** TR alleles collected from all individuals are clustered into TR representatives in terms of sequence similarity. The TR representatives are then classified into TR groups with the same nearest-neighbor SNVs enclosed in parentheses. **d** The frequency distribution (blue) of TR groups with the same nearest SNVs when the minimum difference in length between the longest and shortest TR representatives is set to various thresholds in the *x*-axis. The frequency distribution of TR loci is also shown (orange).

loci. The median ratio was more than 0.9 when the median TR length was <1 kb; otherwise, the median ratio became ~0.8, and the lower quartile ratio was ~0.4 due to a couple of reasons. As the coverage of reads at a locus became smaller, it was more difficult to distinguish two TR alleles separately; only one TR allele was detected, and the other allele was overlooked. Another reason was the difficulty in identifying long TRs with variants and sequencing errors. To analyze the genome-wide characteristics of the majority of TR loci where abundant TR alleles are observed, we hereafter considered TR loci of ratio 0.5 or more (Supplementary Fig. 4b).

### Difficulty of imputing TR representatives using surrounding SNVs

Next, TR alleles from individuals were merged and were clustered as TR representatives for the population. To examine whether the TR representatives had greater genetic divergence than the surrounding SNVs, TR representatives were further clustered into TR groups with identical pairs of nearest SNVs (Fig. 1c). To quantify the divergence of one TR group, the difference in length between the shortest and longest TR representatives can be an indicator. Figure 1d shows the frequency distribution of TR groups with the same nearest SNVs for various minimum difference values. For example, when the minimum was set to 10 bp (100 bp, respectively), 322,382 (16,936) TR groups were found at 200,841 (8621) loci. Overall, these TR groups were difficult to estimate from the surrounding SNVs, indicating that TRs have greater genetic divergence than the surrounding SNVs.

### Measurement of genetic divergence among TRs at each locus

A TR locus may have many TR representatives of different lengths, as illustrated in Fig. 1a, and distinct TR representatives of the same length (Fig. 2a). To resolve the ambiguity, the length of a TR locus is defined as the median length of all TRs at that locus. Figure 2b shows the corresponding length distribution of TR loci.

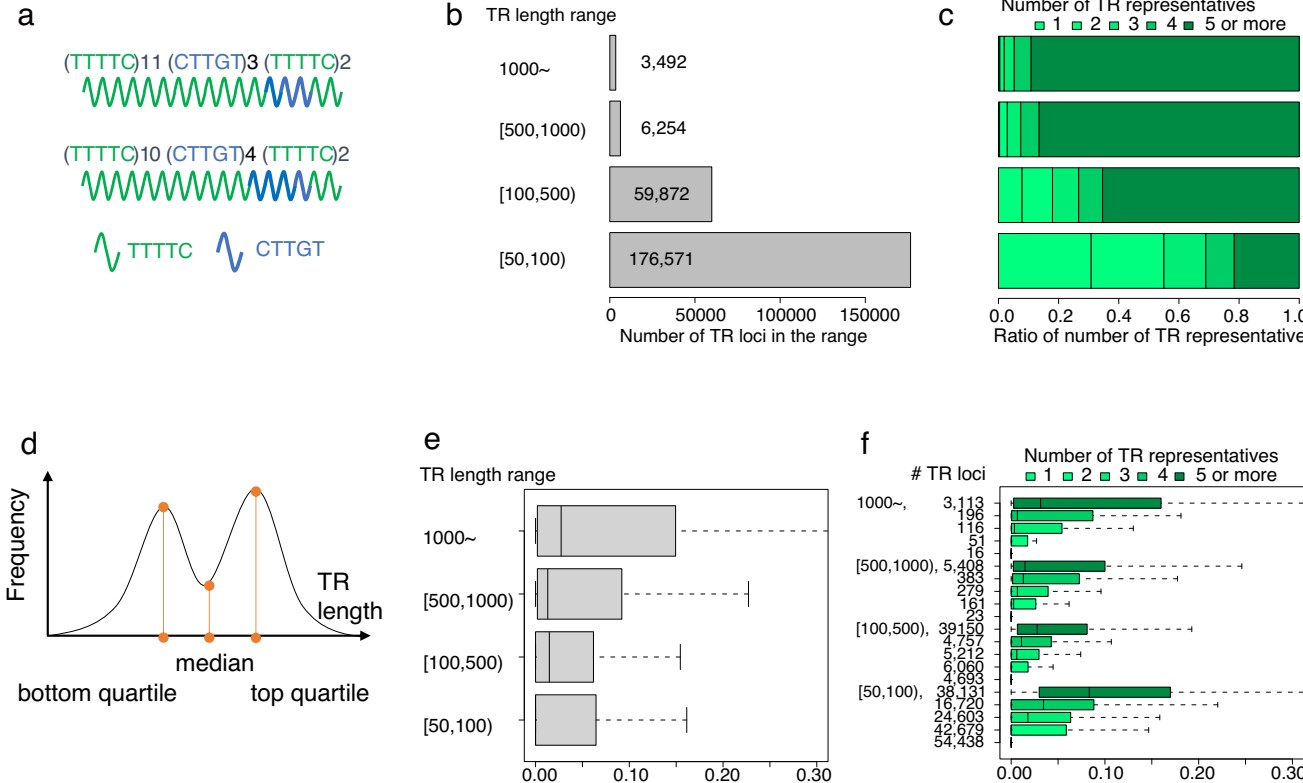

**Fig. 2 | Genetic divergence of TRs measured by number of TR representatives and IQR. a** Examples of different complex repeats of the same length with two different units illustrated by colored waves. **b** Length distribution of TR loci where abundant TR alleles (≥50% of all possible TR alleles) are observed. The length of a TR locus is defined as the median length of TRs at the locus, and TR length ranges (e.g., [100, 500]) are shown on the left. In each length range, the number labeled with the bar indicates the number of TR loci. **c** The proportion of the numbers (1, 2, 3, 4, and 5 or more) of TR representatives in TR loci grouped by length range. Lighter green represents fewer TR representatives. **d** An example of a bimodal frequency distribution. The *y*-axis is the frequency of TRs of the length shown on the *x*-axis. Orange bars indicate the bottom quartile, mean, and top quartile. IQR is the ratio of the difference between the top and bottom quartiles to the median. **e** Boxplots show the IQR distribution of 270 independent individuals for each range, with one boxplot showing the minima, 1st quartile, 2nd quartile (center), 3rd quartile, and maxima. **f** Boxplots show the IQR distribution of 270 independent individuals for each TR length range, with one boxplot showing the minima, 1st quartile, 2nd quartile (center), 3rd quartile, and maxima. TR loci in each TR length range are further partitioned by the numbers of TR representatives, which highlights the significant correlation between two divergence measures, IQR in the *x*-axis and the number of TR representatives in the *y*-axis ($p < 10^{-18}$ in each TR length range in terms of two-sided Spearman's rank test).

To quantify the degree of genetic divergence of a TR locus, the number of TR representatives for that locus is used as an indicator. Accordingly, the genetic divergence of TR loci is shown in Fig. 2c; longer TR loci tended to have more TR representatives and greater genetic divergence.

The length distribution of TR representatives at a TR locus also indicates the genetic divergence of the locus. The length distribution is rarely normal but can be bimodal or multimodal (Fig. 2d); measuring the distribution according to the standard deviation is difficult. Therefore, as a more robust measure, we employed the interquartile range ratio (IQR), which is the ratio of the difference between the top and bottom quartiles to the median. Figure 2e shows the IQR distribution classified by TR locus length, indicating that longer TR loci have greater IQR values. IQR is not equivalent to the number of TR representatives because a locus may have many different TR representatives of equal length (Fig. 2a). However, Fig. 2f shows that TR loci with more TR representatives have larger IQR values across the four ranges of TR locus length, indicating significant correlation between the two measures of genetic divergence ($p < 10^{-18}$ in all four ranges according to two-sided Spearman's rank test). These observations indicate that longer TR loci are generally likely to be more genetically divergent.

**Features related to divergent TRs**

To understand the factors driving TR locus divergence, several features of TR loci were examined. A previous study suggested that in the absence of many mutations, replication slippage occurs frequently, increasing the number of units in tandem; however, when many mutations are introduced, replication slippage is less likely to occur, and TR expansion stops[12]. We confirmed this tendency across the genome. To consider the effect of mutations on divergence, we measured the percentage of difference (i.e., the mutation rate) between a TR and its constituent units (Fig. 3a). This measure can be generalized for complex TRs with multiple units ("Methods"). Because a single TR locus can have different TR representatives, we also extended the measure of mutation rate to TR loci. By treating the frequency of each TR representative in a TR locus as its weight, we calculated the weighted average of mutation rate respectively ("Methods").

Higher divergence in terms of the number of TR representatives was significantly correlated with smaller average mutation rates (Fig. 3b, $p < 10^{-3}$). The same tendency was also observed according to IQR, another divergence measure (Supplementary Fig. 5). Smaller average mutation rates indicate younger replication slippage, and therefore more events of younger replication slippage are involved in the higher divergence of TR loci.

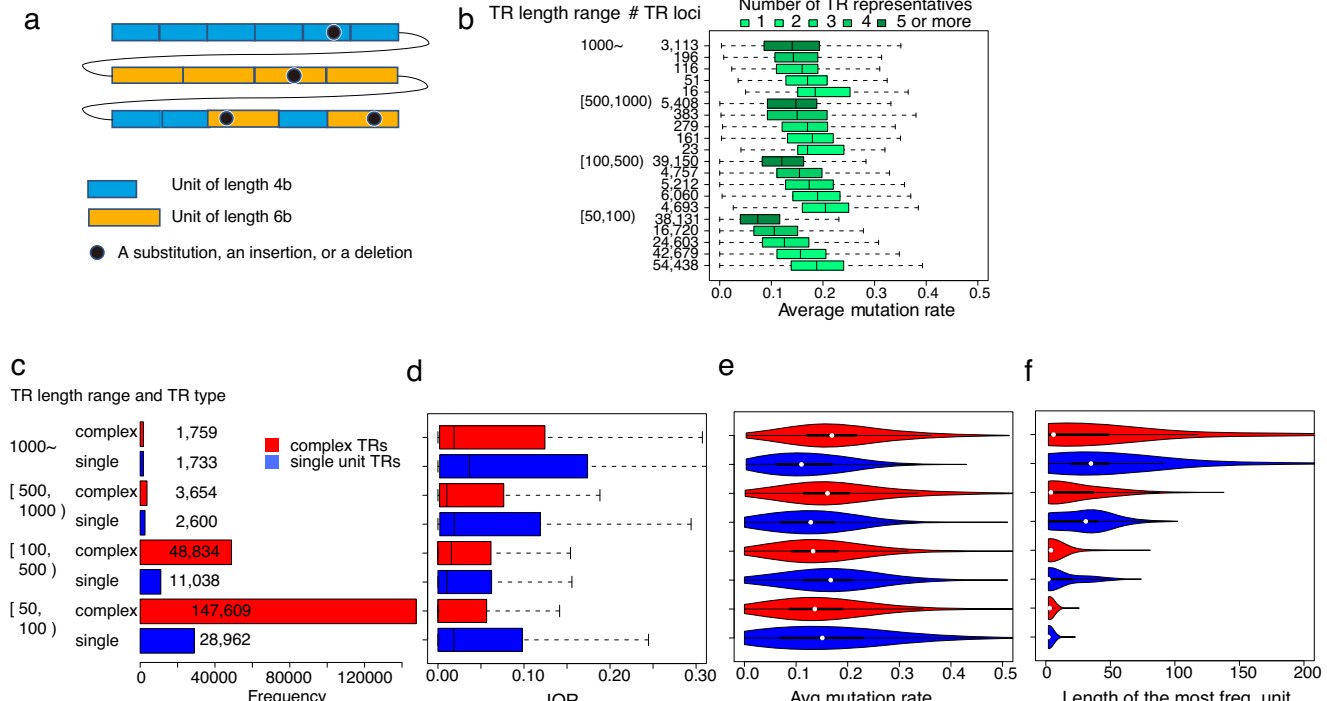

**Fig. 3 | Factors related to TR genetic divergence, and differences between complex and single-unit TRs. a** The figure illustrates a complex TR that has nine blue units of length 4b, and six orange units of length 6b. The top 6 blue units, bottom-left 2 blue units, and middle 4 orange units occur in tandem. Four black dots indicate a substitution, an insertion, or a deletion, and hence the mutation rate is 4/72, where 72 is the total bases in the string. **b** Boxplots show the average mutation rate distribution of 270 independent individuals for each range of number of TR representatives, with one boxplot showing the minima, 1st quartile, 2nd quartile (center), 3rd quartile, and maxima. Lighter green represents fewer TR representatives. The number of TR representatives is significantly negatively correlated with the average mutation rate ($p < 10^{-3}$, two-sided Spearman's rank test).

**c** The frequency distribution of complex TRs (red) and single-unit TRs (blue) in each TR length range. **d–f** Boxplots show the IQR distribution (**d**), average mutation rate distribution (**e**), and length distribution of the most frequent units (**f**) of 270 independent individuals for each class of single-unit TRs (colored blue) and complex TRs (red). Each boxplot shows the minima, 1st quartile, 2nd quartile (center), 3rd quartile, and maxima. When TR length is 500 or more, the IQR distribution (**d**), average mutation rate distribution (**e**), and unit length distribution (**f**) differ significantly between complex TRs and single-unit TRs, and their $p$-values are smaller than $10^{-4}$ (**d**), $10^{-88}$ (**e**), and $10^{-24}$ (**f**) respectively (in terms of two-sided Spearman's rank test).

## Features that highlight the differences between complex and single-unit TRs

Figure 3b was observed after mixing two different situations when a single unit is dominant at a TR locus and when multiple units are present within a complex TR. Because this study performs genome-wide analysis of complex TR loci, these two situations are examined separately. We found that complex TRs are more than the single-unit TRs in loci (Fig. 3c), highlighting the importance of complex TRs. The complex TR loci with ≥500 bp TRs showed significantly lower genetic divergence in terms of lower IQR (Fig. 3d, $p < 10^{-4}$) and a larger mutation rate (Fig. 3e, $p < 10^{-88}$) than the single-unit TR loci did, demonstrating the general trend of lower (higher, respectively) divergence at higher (lower) mutation rates. We also revealed a tendency for complex TRs to consist of <10-bp units and single-unit TRs to be minisatellites with a statistical significance (Fig. 3f, $p < 10^{-24}$).

## TR loci with markedly expanded TRs as candidates for disease association

Approximately 60 diseases are known to be associated with rare disease-specific TRs that are markedly elongated compared with TRs in normal control samples[2]. Although control samples were used in this study, it would be meaningful to create a genome-wide collection of TR loci with extremely expanded TR representatives to understand the characteristics of this extreme expansion. Specifically, as one indicator of such TR loci, we used the length difference between the longest and median TRs, and we observed the unit length distribution of the most frequent units in the longest TRs (Fig. 4a). When the difference was

>300 bp, the median unit length of the longest TRs was >15 bp, indicating that more than half of the longest TRs were minisatellites.

Expanded minisatellites have been recently reported to correlate with ALS[22], Alzheimer's disease[21], and schizophrenia and bipolar disorder[20]. ALS risk was found to be significantly associated with 69-mer copy expansion at chr18:57024495-57024955 (hg38) through a comparison of cases and control samples[22]. Notably, the number of repeated copies in the Japanese control samples [median(IQR) = 23.6(15.6−28.6), mean ± standard deviation (SD) = 22.3 ± 9.63] was much larger than that in the ALS samples [median (IQR) = 17.5(9−24), mean ± SD = 17.7 ± 10.4] that were of European descent[22] (Fig. 4b). Bipolar disorder and schizophrenia risk was associated with several mutations in the 30-mer unit of chr12:2255791-2256090 (hg38)[20]; three risk unit variants were found in 34 of 540 alleles in the Japanese control samples (Supplementary Fig. 6). Together, minisatellite copy number varied markedly among control samples from two different populations, suggesting careful development of different disease models from minisatellites for different populations.

Although minisatellites are prevalent, markedly long TRs containing short units (<10 bp) were also observed (Fig. 4a). It is noteworthy that marked expansion of disease-specific short units, different from the most frequent units in control samples at disease-associated loci, has recently been reported in nine diseases[2]. For example, among patients with CANVAS, AAGGG is copied 400−2000 times in late-onset ataxia, whereas AAAAG is usually copied most (11 times) in controls[25], showing that the locus has complex TRs (see TRs in Group II of Fig. 1a).

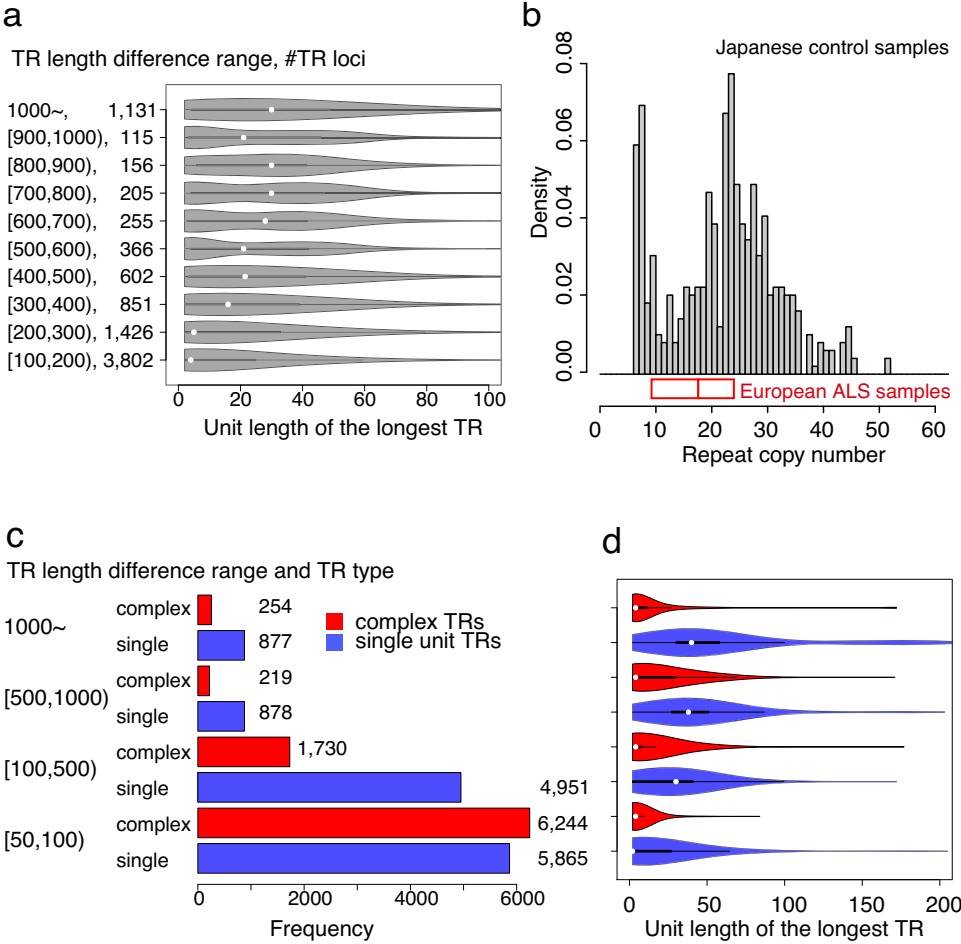

**Fig. 4 | Analysis of the longest TRs as disease-associated TRs and characterization of their generating units. a** The unit length distribution of the most frequent units of the longest TRs such that the TR length difference between the longest and median TRs is in the range shown at left. The second left shows the numbers of TR loci. The boxplot inside each violin plot shows the minima, 1st quartile, 2nd quartile (center), 3rd quartile, and maxima. **b** Analysis of expanded copies of the 69-mer unit at chr18:57024495-57024955 in the human reference genome (hg38) that are correlated with ALS samples of European descent. The red boxplot below the histogram shows the 1st, 2nd, and 3rd quartiles of copy number of the 69-mer unit in the ALS samples. The histogram shows the density distribution of the copy number of the 69-mer unit in Japanese control samples ($n = 270$). **c** TR loci are classified according to whether the longest TR representative is complex TR (red) or single-unit TR (blue). Single-unit longest TRs are more than complex ones when the length difference is 100 or more. **d** The unit length distribution of the most frequent key unit in the longest TR representative of 270 independent individuals. The boxplot inside each violin plot shows the minima, 1st quartile, 2nd quartile (center), 3rd quartile, and maxima. Similar to Fig. 3f, units in complex TRs were significantly shorter than single units ($p < 10^{-11}$ according to two-sided Spearman's rank test) when the TR length difference was >100 b.

To understand the prevalence of this type of expansion, TR loci with extremely expansion are categorized into two groups depending on whether the longest TR consists of a single dominant unit or complex multiple units ("Methods"). Figure 4c shows that 2203 (24.7%) TR loci had complex units among 8909 in which the longest TRs were >100 bp longer than the median, and their most frequent units are significantly shorter than those in single-unit TRs ($p < 10^{-11}$, Fig. 4d). These complex TR loci are potential candidates for disease-associated regions. Overall, our genome-wide collection of TR loci can be a resource for searching for disease-associated minisatellites and short tandem repeats (Supplementary Data 1).

We used our complex TRs to augment the maintained gnomAD table of 60 known disease-associated TR loci generated from short-read sequencing data in ExpansionHunter because ExpansionHunter is not designed to detect complex tandem repeat patterns in long leads. Specifically, the TR patterns observed in this study are added to each of the 60 entries (Supplementary Data 2, "Methods"). For example, in the *AFF2* gene, pattern $(GT)_{12}(CCG)_{56}(AGCC)_5(CCG)_9$ is frequently observed in our study, but only CGG-repeats are registered in gnomAD (see Supplementary Data 2), demonstrating that our study can

complement complex tandem repeats missing in gnomAD. According to the known rules for testing whether a TR is pathogenic or not, some individuals had dominant alleles with expanded repeats in the *ATX-N8OS* gene coding regions (Supplementary Data 2), suggesting the individuals are carriers. Stevanovski et al. examined a cohort of 37 individuals with 25 neurogenetic diseases and identified repeat expansions with 500 or more repeat units in *C9orf72, DAB1, DMPK*, and *FXN*[33]. These expansions were pathogenic and are not observed in our control samples (Supplementary Data 2). In *RFC1*, they reported $(AAGGG)_{>500}$ repeat expansions, while we found $(ACAGG)_{>500}$ repeat expansions, in which different units are expanded. These differences may be seen because they and we used case and control samples, respectively.

**Phylogenic analysis of divergent TRs at each locus**
To track the evolution of high TR diversity within a population, it is informative to create a phylogenetic tree. Such trees are distinct from conventional phylogenetic trees constructed based on single-nucleotide mutations; within TRs, there is a higher frequency of repeat unit duplication and contraction compared to adjacent SNVs

(Fig. 1d). Therefore, when considering the inter-sequence distances of a phylogenetic tree, unit-by-unit duplications and contractions must be considered in addition to single-nucleotide mutations, insertions, and deletions. We implemented an algorithm to draw such a phylogenetic tree ("Methods"). Figure 5 shows a phylogenetic tree of TR representatives in an intron of *RFC1* at chr4:39,348,424-39,348,485 in the reference genome (hg38), and TRs are largely categorized into two groups, as illustrated in Fig. 1a. One group has simple TRs, (e.g., $(ACAGG)_i$) as well as complex tandem repeats (e.g., $(AAAGGG)_j(AAAGG)_k$, $(AAAGG)_i$ $(AAAGGG)_j(AAAGG)_k$) and share the same downstream nearest SNV at chr4:39,348,321 in common. By contrast, the other group has only simple repeats of the form $(AAAAG)_i$, where *i* ranges from 9 to 12, and the surrounding 1 kb regions are consistent with the reference genome. Supplementary Fig. 7 presents another example of the phylogenetic tree of TRs in an intron of *CNBP* at chr3:129,172,576-129,172,656. These examples illustrate the general

trend that unit extensions diverge much faster than their surrounding regions (Fig. 1d).

## Discussion

To facilitate the analysis of complex TRs composed of different units, we used a mathematical model and accurate algorithms for computing the model that we developed[30] and examined highly accurate, long reads (median of -14 kb) from 270 Japanese control samples. The results provided a landscape of complex TRs hidden within individual human genomes. Of note, -322 k TRs were difficult to estimate (impute) from the surrounding SNVs. Complex TRs are more than single-unit TRs. High genetic divergence in TRs was correlated with low mutation rates, suggesting the active involvement of recent replication slippage. We observed a statistically significant tendency for complex TRs to consist of <10-bp units and single-unit TRs to be minisatellites at loci with ≥500-bp TRs. This study also provides insight into extended TRs associated with disease. Among 8909 loci with TRs longer than

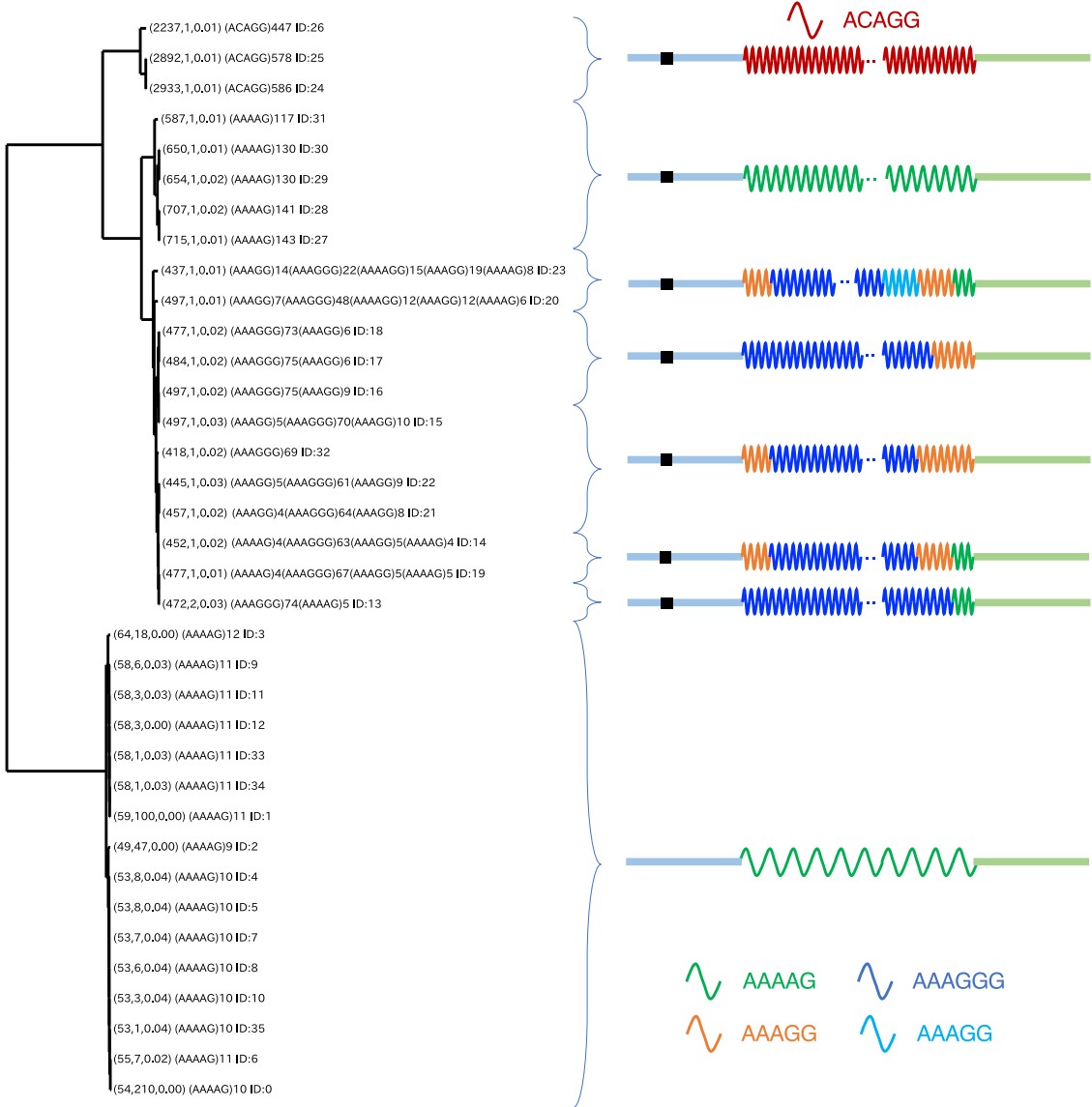

**Fig. 5 | Phylogenic analysis of divergent TRs.** The left phylogenetic tree shows the evolution of complex tandem repeats in an intron of *RFC1* at chr4:39,348,424-39,348,485 in the human reference genome (hg38). The right illustrates several tandem repeat patterns with different units represented by colored waves. The black box to the left of the tandem repeat indicates the nearest SNV at chr4: 39,348,321, which is common to the upper seven tandem repeat patterns. In the left tree, the first three numbers of each TR representative, for example (54, 210, 0.00) in the bottom, show the length of the TR representative, the number of TR alleles in the TR representative, and the discrepancy rate between the decomposition and TR representative.

100 bp above the median detected in this study, ~75% were single-unit TRs and were often minisatellites such as those associated with bipolar disorder, schizophrenia, Alzheimer's disease, and ALS. The remaining ~25% were complex TRs consisting mainly of <10-bp units such as CANVAS and BAFME. In the literature, family-centric approaches have made it possible to detect most of the approximately 60 diseases correlated with markedly expanded TRs to date[2]. To complement the classic pedigree method, we hope that our database of TRs will offer a unique approach to narrowing down the regions of relevance to human disease.

Several intriguing research questions remain unexplored. In this present study, we discussed TR diversity in the Japanese population. The results show that TR diversity is large even in genetically small Japanese populations. We hope to apply our methods to the entire human population in a future study.

Here are some caveats and limitations regarding the study design. Because the DNA used for long-read sequencing in this study was extracted from immortalized B cells, it is possible that specific TR alleles were selected during the EBV-induced immortalization process and did not necessarily represent the original allele. Generation of fully haplotype-resolved diploid human genome assemblies is resource-intensive, requiring 30- to 40-fold coverage of HiFi reads from single individuals and high coverage of Hi-C and Nanopore sequencing data. We may need >1000-fold coverage of HiFi reads to address somatic variability in TRs[34]. Therefore, there is uncertainty as to whether the sequence reported in this study is the predominant repetitive structure or just one of the other structures that may be present at these loci in these individuals. In this study, to acquire data from a larger number of individuals, one single-molecule real-time sequencing cell (SMRT cell) was used to obtain moderate coverage (≥7.5) reads, which, in theory, allowed reliable observation of the majority of complex TRs ("Methods", Supplementary Fig. 2). To collect more information about the Y chromosome, 258 of the 270 control samples are male. This design halves the number of alleles on the X chromosome compared to autosomes, but allows the collection of approximately the same number of alleles on the Y chromosome, thereby detecting 41 complex TR loci in which the longest TRs were >100 bp longer than the median (Supplementary Data 1).

The pathophysiological mechanism of TR involvement in a variety of diseases is a primary question. The expansion of CGGs in promoters has been reported to result in changes in methylation[35], translation initiation other than ATG[36], and changes in the position of DNA-binding proteins[37], which can affect the expression levels of downstream genes. (CAG)-repeats and (GCG)-repeats within the protein-coding region can produce toxic proteins with unusual folded structures[2]. Complex TRs in introns can alter splicing and induce transcription abortion[29,38]. We speculate that complex TRs in introns or in intergenic regions may influence promoter-enhancer interactions during brain development[39] because CTCF loops, the basic chromatin structures of the nucleus[40], have been reported to be established at gastrulation[41]. Therefore, it would be meaningful to understand whether CTCF loops vary around the expanded TRs reported in this study.

Another open question is the method for constructing a useful database of diverse TRs. SNV databases such as GnomAD[42] describe the frequency of major and minor alleles and their genotypes and are essential for conducting genome-wide association studies and assessing the frequency of SNVs found in exome/whole genome resequencing. However, TRs are significantly more diverse than SNVs. As a step toward describing the genetic divergence of TRs at each TR locus, we propose a method for classifying TR alleles from individuals in terms of global alignment edit distance to produce TR representatives for the population. Because each TR representative is simply a nucleotide sequence, we propose a method to show each as a TR by breaking it down into one or more units. When mutation rates become large in the longer units, it becomes crucial to analyze minisatellites

associated with a disease because a representative long unit can have several unit variants that are specifically related to the focal disease. Indeed, two recent noteworthy studies have provided detailed examinations of long-unit variants associated with ALS[22] and bipolar disorder and schizophrenia[20]. Therefore, an approach that automatically detects and evaluates long-unit variants genome-wide would be useful. To that end, the TR patterns identified at the TR locus in this study would be useful inputs to TR-specific genotyping tools such as TRGT[43]. Collectively, it would be quite meaningful to design a database by considering the characteristics of each TR locus, including complex TRs with short units, minisatellites with long-unit variants, and longest expansions with units that differ from the most frequent units.

## Methods

### Ethics statement

This study was approved by the Research Ethics Committee of the Faculty of Medicine of the University of Tokyo (Human Genome/Gene Analysis Research Ethics Review; review no. 19-323).

### DNA sample preparation

In this study, we used immortalized B cells derived from Japanese subjects that were distributed by the Japanese Collection of Research Bioresources (Japanese B cell DNA bank), the National Institute of Biomedical Innovation, Health and Nutrition. To collect more information about the Y chromosome, 258 of the 270 control samples are male. For SMRTbell library preparation, B cell DNA was sheared twice using a Diagenode's Megaruptor 2 (Diagenode, Denville, NJ, USA) set to 25 kb, and purified using a 1× volume ratio of AMPure PB beads (Pacific Biosciences, Menlo Park, CA, USA). DNA sizing was checked on the FEMTO Pulse (Agilent) using the Genomic DNA 165 kb kit on extended mode. SMRTbell libraries for sequencing were prepared using the Procedure & Checklist−Preparing HiFi SMRTbell Libraries using the SMRTbell Express Template Prep Kit 2.0 protocol. Briefly, the steps included DNA repair, overhang adapter ligation using the SMRTbell Express Template Prep Kit 2.0 (Pacific Biosciences), 10-kb cutoff size selection using the BluePippin DNA Size Selection System by Sage Science, and binding to polymerase using the Sequel II Binding Kit 2.2 (Pacific Biosciences). Sequel II CCS/HiFi libraries were sequenced using the Sequel II Sequencing Plate 2.0 and SMRT Cells 8M (Pacific Biosciences) with a movie length of 30 h.

### Probability of detecting SVs from long reads

We considered the probability of detecting a focal structural variant (SV) of size $d$ from a set of $N$ PacBio HiFi reads of length $L (\geq d)$ that are sequenced from a given genome of $G$ in size. First, we calculated the probability of failing to detect the focal SV using the idea of Lander-Waterman statistics. If the focal SV starts from position $x$, its range is $[x, x + d)$, which represents the half-open interval from $x$ to $x + d - 1$. Supplementary Fig. 2a shows an example in which no reads of length $L$ cover the range $[x, x + d)$, which is equivalent to having no reads start from any position within the range $[x + d - L, x]$. The probability of such an occurrence was approximated using the following formula, assuming that reads start from any position with an identical probability $N/G$:

$$\left(1 - \frac{N}{G}\right)^{L-d+1} = \left(1 + \left(-\frac{N}{G}\right)\right)^{\frac{1}{-\frac{N}{G}} \cdot -\frac{LN}{G} \cdot \frac{L-d+1}{L}} \approx e^{-\frac{LN}{G} \cdot \frac{L-d+1}{L}} = e^{-c\left(1-\frac{d-1}{L}\right)} \quad (1)$$

where $c$ in the last term denotes $LN/G$, which is the genome coverage by reads. The last approximation in Eq. (1) was obtained by assuming that $N/G$ is close to 0. Supplementary Fig. 2b shows that the probability ($y$-axis) of covering the focal SV depends on the values of coverage $c$ ($x$-axis), the SV length ($d$), and the read length ($L$). The probability increased as we detected shorter SVs, used longer reads, and/or had reads of higher genome coverage. For example, the value

of $L$ has little influence in detecting SVs that are 1 or 2 kb in length. To list more SVs longer reads and/or higher genome coverage are required. To detect SVs present in one of the two alleles of a diploid genome, the genome coverage must be halved. For example, the probability of finding an SV of length 5Kb (1Kb, 2Kb, respectively) in one allele from a set of reads of length 14Kb is 91.02% (95.98%, 96.92%) when the genome read coverage is 7.5 (3.75 per allele).

## Identification of TRs in the reference human genome and in individual reads

We searched the reference genome (or single reads) for non-overlapping regions with complex TRs using mTR, which detects short TRs and minisatellites of ≥10 bp units with high sensitivity[24]. When mTR was too sensitive to divide complex TRs with short units into small regions, we also used RepeatMasker, which was likely to output longer contiguous regions than mTR. We generated two separate lists of non-overlapping regions using mTR and Repeat-Masker and merged them into a single list (denoted by $L$) that may include overlapping regions. To generate the final list (denoted by $F$) of non-overlapping regions from $L$, we repeated the process that first merged overlapping regions in $L$ into non-overlapping regions and then merged non-overlapping regions within a distance of 10 bp to output the final list $F$. More than half of all TRs above 500 bp were detectable using mTR alone (Supplementary Fig. 3b); most of them are minisatellites (Supplementary Fig. 3c). We then aligned reads with TRs to the reference genome using minimap2 to locate the TR.

## Comparison between mTR and TRF for detecting minisatellites

Because TRF likely outputs multiple overlapping tandem repeats at the same locus, whereas mTR outputs non-overlapping TRs, it was ambiguous to uniquely compare the TRs of mTR and TRF at each locus. To avoid this ambiguity, we generated non-overlapping TR regions from overlapping TRs output by TRF. Precisely, we merged TRs of TRF that were overlapping or within a distance of 10 bp into non-overlapping TR regions and associated the longest TR unit in the merged TRs with each non-overlapping TR region. The length and unit length of a pair of TRs output by mTR and TRF at each locus were compared, and their length distributions are shown in Supplementary Fig. 3d, e.

## Computing TR alleles in an individual genome

A TR at each locus in one individual is derived from one of two homologous chromosomes. Therefore, TRs from the identical homologous chromosome should match each other almost perfectly in the sense of an optimal global alignment that maximizes matches from the beginning to the end of a TR. Specifically, we avoided local alignment, which can extract a highly similar but short subsequence between two given sequences. To calculate an optimal global alignment between a pair of biological sequences such as TRs, we used the KSW2 library[44,45]. Using this method, we constructed the edit distance matrix of the global alignments between all pairs of TRs at each locus in an individual human genome.

Using the matrix, we clustered TRs into one or two groups representing homozygosity or heterozygosity of the TR locus using the neighbor-joining method[46], which is a widely used efficient heuristic algorithm for clustering DNA strings, according to the edit distance matrix. Specifically, we imposed the condition that the diameter of one group (the maximum edit distance between any pair of TRs) was at most 1% of the longest read length in the group; and the group was defined as valid if this condition was satisfied. The threshold was set to 1% because a >1% discrepancy between the two alleles was assumed, though the threshold can be also set to a different value. As the representative allele of one group, we selected the centroid, which minimizes the sum of distances to all members in the group. Hereafter,

we refer to single- or two-centroid TRs as TR alleles. The program used for this analysis is available at: https://github.com/morisUtokyo/cTR.

## Computing representative TRs for the Japanese population

We collected TR alleles from 270 Japanese individuals as described above, divided the alleles into valid groups as defined above using the neighbor-joining method according to the edit distance matrix between all TR alleles, and selected the centroid of each valid group as a TR representative for the Japanese population. Because this clustering problem is generally intractable (NP-complete)[47,48], we implemented a heuristic algorithm. We define that an internal node $x$ in the neighbor-joining tree as valid if the set of all leaves in the subtree rooted at $x$ is valid (i.e., a valid node represents a valid group), and $x$ is defined as maximally valid if any of its ancestors is not valid. We repeated the heuristic process for selecting a maximally valid internal node until the root of the neighbor-joining tree became valid, output the group of leaves in the subtree rooted at the node, and removed the node and its subtree from the neighbor-joining tree. After removing the subtree, we updated the diameter of each internal node. The program used in this analysis is available at: https://github.com/morisUtokyo/cTR.

## Algorithm that selects repeat units in a complex TR and decomposes the TR into units

To automatically characterize the sequence configurations of complex TRs with multiple repeat units, we propose how to measure the goodness of selecting a set of repeat units denoted by $U$. Once $U$ is selected, it is tractable to compute an optimal concatenation of units that partially matches the TR with the minimum Levenshtein distance (i.e., the sum of substitutions, insertions, and deletions) in the presence of sequencing errors using an efficient algorithm for solving the approximate regular expression matching problem[49] (This concept has been recently reinvented in the string decomposer algorithm[50]). We developed an algorithm for selecting nearly optimal set of repeat units. The details of the computational complexity analysis, implementation, and experimental results of our program are found in our companion paper[30]. Our program is available at: https://github.com/morisUtokyo/uTR. The algorithm is outlined below.

To define the measure of selecting a better unit set, let us consider how to decompose an input TR (denoted by $S$) into a series of neighboring substrings that are present in $U$, which is called a decomposition $D$ of $S$ by $U$. Assuming maximum parsimony that prefers fewer events of replication slippage and/or non-homologous recombination, a better decomposition should consist of fewer copies of fewer and shorter generating units. To achieve this intuitive goal, we define the penalty of $D$ by $U$ to be smaller when $D$ is a better decomposition. Specifically, we formally define the penalty of unit $u$ as the sum of its length, $|u|$, and the number of its occurrences, $occ(u)$, in $S$; i.e., $|u| + occ(u)$. The penalty of $D$ by $U$ is then defined as the sum of penalties of all repeat units, $\sum_{u \in U} |u| + occ(u)$.

In practice, we also need to accommodate sequencing errors that often generate infrequent units and may unnecessarily enlarge the unit set $U$. To exclude these rare units so that $U$ can have essential units only, we measure the degree that most substrings in decomposition $D$ are also present in $U$, and we define the coverage of decomposition $D$ by $U$ as $\sum_{s \in D, s \in U} |s|$. Thus, it is ideal to find $U$ that minimizes the penalty of $D$ by $U$ and maximizes the coverage of $D$ by $U$. However, it may not be possible to optimize both of these criteria simultaneously. Of note, as the latter maximization is equivalent to the minimization of $\sum_{s \in D, s \notin U} |s| (= |S| - \sum_{s \in D, s \in U} |s|)$, we attempt to minimize the penalty by redefining it as follows:

$$\sum_{u \in U}(|u| + occ(u)) + \sum_{s \in D, s \notin U} |s| \qquad (2)$$

It is an open question whether it is tractable to compute a pair of $U$ and $D$ that minimizes the above redefined penalty (2). Because of this situation, we implemented a greedy algorithm that repeats the process of selecting and adding to $U$ the best unit that minimizes the above redefined penalty.

Using synthetic benchmark data, we showed the practical feasibility of detecting typical complex TRs in human genomes with almost 100% accuracy as long as the sequencing error rate is less than 1% (which can be assumed for the PacBio HiFi sequencer) and the computational efficiency of processing complex TRs in time linear to length.

### Mutation rate of a TR and a TR locus

The mutation rate of a TR is defined as the ratio of the Levenshtein distance between the TR and its optimal concatenation of units to the TR length. For TR loci with multiple TR representatives, the average mutation rate weighted by the frequency of each TR representative is defined as the mutation rate of the TR locus.

### Finding reliable SNVs surrounding each TR

Rather than using publicly available SNV databases, we attempted to find SNVs among HiFi reads from scratch because we intended to identify rare SNVs that might be associated with rare TRs. To detect SNVs that differ from the reference hg38 genome in the 1-kbp genomic sequences around each TR, these 1-kbp sequences were mapped to the reference hg38 genome using the minimap2 program[44]. To retrieve SNVs from the alignments, we parsed the cs SAM/PAF tags of the surrounding 1-kbp sequences and identified reliable SNVs that were not sequencing errors according to a statistical test. Although the average sequencing error of HiFi reads is quite small, at approximately 0.1%[51], we observed errors such as single nucleotides and indels at random positions because we handled about 10-fold 1-kbp sequences from each of ~300 individuals. Substitution errors are less frequent than indel errors among HiFi reads[51]. Therefore, we ignored indels but retained significant substitutions, with $k$ occurrences in $n$ reads, such that the probability of observing $k$ or more occurrences was lower than the 5% significance level, assuming that a substitution was observed at random with probability $p$ (e.g., 0.05%, half of the average sequencing error). We treated these significant substitutions as reliable SNVs but treated the other substitutions as sequencing errors. For example, when $n = 3000$ and $p = 0.05\%$, we set $k = 4$. The program used in this analysis is available at: https://github.com/morisUtokyo/hTR.

### TR representatives and TR loci

Following decomposition of a TR representative, units are ordered according to their occurrence frequency in the decomposition, and the most frequent $k$ units, such that 90% or more bases occur in the $k$ units are calculated and are called key units. If $k = 1$, the TR representative is treated as having one unit; otherwise, if $k \geq 2$, it has $k$ multiple units and is defined as complex.

When a TR locus has one TR representative, it is straightforward to establish that the locus has a single unit (or multiple units), according to the status of the representative does so. When the TR locus has more than one TR representative, checking whether the set of key units in all TR representatives has a single unit or multiple complex units, we determine the status of the TR locus. Notably, even if one TR representative is complex, the entire TR locus is also defined as complex.

### Computing phylogenetic trees considering expansion and contraction of TRs

To create a phylogenetic tree of TR representatives that could track the evolution of high TR diversity in the population, the distance matrix between all pairs of TR representatives was calculated, and subsequently, a phylogenetic tree was created from the distance matrix using the nearest-neighbor join method[46]. Computing the distance matrix is crucial because, in addition to single-nucleotide mutations, insertions, and deletions, unit duplications and contractions must be considered. This is called the edit distance with duplication and contraction (EDDC) problem, and polynomial time algorithms are proposed for solving the problem[52]. We implemented one of the algorithms that runs in time $O((|\Sigma| + |U|)(n^3 + n^2 u^2))$, which means that the longest computational time is proportional to $(|\Sigma| + |U|)(n^3 + n^2 u^2)$, where $\Sigma$ denotes the set of letters used, $n$ is the length of the input string, and $u$ is the maximum length of units in $U$. The algorithm has the flexibility to define various distance scores (penalties) for match, mismatch, insertion, deletion, unit duplication, and unit contraction. The values set for these parameters, especially the penalties for unit duplication and unit reduction, should be carefully considered, though substantial consideration for this remains to be done. In this study, we tentatively assigned $0, +1, +1, +m/2$, and $+m/2$ to the respective penalties of the six parameters, where $m$ denotes the length of the unit, because a longer unit is less likely to be observed. Fine-tuning of parameter settings will be necessary after more data is collected in the future.

### Annotation of TR loci with extended tandem repeats

Supplementary Data 1 shows TR loci such that the longest TRs were >100 bp longer than the median. If a TR locus is in a gene coding region, we annotated it with the gene name and its location within the gene (exon, intro, UTR, etc.) using the UCSC hg38 tables: https://hgdownload.soe.ucsc.edu/goldenPath/hg38/database/kgXref.txt.gz and https://hgdownload.soe.ucsc.edu/goldenPath/hg38/bigZips/genes/hg38.knownGene.gtf.gz.

We used geneSymbol in the first table above if it was present and spID in the second table otherwise. When a TR overlaps with untranslated regions (UTR) and exons, it is treated as being associated with UTR. When TR overlaps with an exon but does not have any UTR, it is associated with the exon. When a TR is properly included in an intron, it is labeled with the intron.

### Annotation of 60 disease-associated TR loci

We downloaded the following well-annotated GnomAD table of 60 disease-associated TR regions that were generated from short-read sequencing data by ExpansionHunter: https://gnomad.broadinstitute.org/short-tandem-repeats?dataset=gnomad_r3.

To reinforce the above information, the TR patterns generated in this study were added to each of the 60 entries, and the result is found in Supplementary Data 2. Some repeats appeared pathogenic or abnormal but actually had interruptions and were not pathogenic. For example, (GCA)-repeats in gene *AR* are pathogenic and associated with the spinal and bulbar muscular atrophy of Kennedy when the copy number is 40 or greater. In our control samples, the (GCA)-repeat pattern in some samples appears to have more than 40 copies, e.g., (GCA)54, but impure in the sense that it actually has interruptions and (GCA) has less than 40 consecutive occurrences in the underlying DNA sequence (see Supplementary Data 2). Thus, TRs can be written using simpler impure TR patterns with many mismatches, or more complex pure TR patterns with few mismatches. This is known as Occam's Razor and is a trade-off between the two TR patterns. To point out this problem, in Supplementary Data 2, we annotated some repeats with simple and complex patterns using our program uTR. TR patterns that should be described by simple impure patterns and more complex pure patterns are highlighted in orange in the H column of Supplementary Data 2.

### Reporting summary

Further information on research design is available in the Nature Portfolio Reporting Summary linked to this article.

## Data availability

All sequencing data and TR loci for 270 Japanese samples are deposited in the NBDC Human Database under Data Set ID JGAS000286 and JGAS000505 and are available under restricted access for the preservation of the confidentiality of personal data. Access can be obtained by a direct application for using NBDC Human Data (see the details at https://humandbs.biosciencedbc.jp/en/data-use). As the reference human genome, hg38 was used. We used the gene name and its location within the gene (exon, intro, UTR, etc.) using the UCSC hg38 tables: https://hgdownload.soe.ucsc.edu/goldenPath/hg38/database/kgXref. txt.gz and https://hgdownload.soe.ucsc.edu/goldenPath/hg38/bigZips/ genes/hg38.knownGene.gtf.gz. We used the GnomAD table of 60 disease-associated TR regions: https://gnomad.broadinstitute.org/ short-tandem-repeats?dataset=gnomad_r3. All other data supporting the findings described in this manuscript are available in the article and its Supplementary Information files, and from the corresponding author upon request.

## Code availability

Codes are available at https://github.com/morisUtokyo/cTR (https:// doi.org/10.5281/zenodo.8207183), https://github.com/morisUtokyo/ uTR (https://doi.org/10.5281/zenodo.8207190), and https://github. com/morisUtokyo/hTR (https://doi.org/10.5281/zenodo.8207188). Supplementary Fig. 8 is a flowchart outlining how these programs are used in the various analysis stages.

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

## Acknowledgements

The authors would like to thank Bansho Masutani for analyzing the computational complexity of computing an optimal pair of a unit set and a decomposition, Chie Owa, Haruka Kobayashi, Wei Qu, and Yoko Saito for sequencing Japanese control samples, and Yuta Suzuki for base calling reads. This work was supported in part by the Japan Agency for Medical Research and Development (AMED) [21tm0424219h0001] (S.M.).

## Author contributions

S.M. and K.I. designed the study. T.A. and R.K. developed a program for computing phylogenetic tree considering expansion and contraction of units, and S.M. developed the other programs. K.I. used parallel computers to calculate the results. S.M. analyzed the results and wrote the paper.

## Competing interests

The authors declare no competing interests.
