## [Peer Review File · Nature Communications]

A landscape of complex tandem repeats within individual human genomesReviewers' comments:

Reviewer #1 (Remarks to the Author):

The authors of this study have analysed the PacBio HiFi data of 270 Japanese controls and have created a catalog of mosaic TRs. The findings of this study are very interesting and illuminate an important, yet unexplored feature of TRs. As the authors note, interrupting motifs in TR sequences have significant clinical relevance (e.g., the loss of CAA and CCA interruptions in the HTT gene can cause earlier onset of Huntington disease). Although software like ExpansionHunter (<https://github.com/Illumina/ExpansionHunter>) and REViewer (<https://github.com/Illumina/REViewer>) can characterize the motif composition of TRs and identify the presence of interrupting repeat motifs in the standard Illumina short-read sequencing data, respectively, the read length poses significant challenges in interrogating long TRs and minisatellites and their full-length motif structures genome-wide. The sequencing accuracy and read length of PacBio HiFi make it an ideal technology to study the length and motif composition of complex TR sequences. The authors have created a database of mosaic TRs (8,820 loci) that they highlight could be a valuable resource to study disease associations.

This is a very well-written manuscript with clearly presented data.

1. The database of 8,820 TR loci in Extended data table 1 could include additional annotations (gene names, location within the gene (intron, exon, UTR, etc.), any known disease associations, and/or overlaps with regulatory elements). This would help us better understand the diversity of TR alleles within different genomic contexts.

2. Although this study focuses on control samples, it would still be extremely valuable to characterize the allelic diversity and motif compositions of the 60 known disease-associated TR loci that may not harbor extreme expansions. Stevanovski et al. 2022 (PMID: 35245110) reported diverse TR alleles with pentanucleotide motifs and identified several unique repeat motifs in their analysis of clinically nonaffected individuals. I suggest including the TR sizes and motif compositions of the 60 known disease loci of this cohort. For normal TR alleles that can be spanned entirely by reads, ExpansionHunter calls from short reads are quite reliable. And gnomAD database includes the ExpansionHunter genotype calls as well as the read pileups of the disease TRs (https://gnomad.broadinstitute.org/short-tandem-repeats?dataset=gnomad_r3), and this could be a useful resource for comparing the observations in this cohort vs other populations. And perhaps the authors could also mention the gnomAD resource for disease TR loci in their discussion.

3. Have the authors used other TR-specific genotyping tools to characterize the expanded TRs in their dataset, or as a confirmation of their repeat size calls? For example, TRGT

(<https://github.com/pacificBiosciences/trgt/>). Could the authors comment on the general strategies they had adopted to confirm or verify their genotype calls?

4. It would be really interesting to look at the methylation profiles of the expanded TRs the authors have identified in this study. Have the authors looked at the methylation characteristics of the expanded TRs? I suggest including this data.

5. The authors list the TR candidates in the hg38 reference genome. Could the authors remark on the TR loci and their motif compositions in the hg38 vs the T2T reference sequence?

6. Fig 2c. Revise “5<=” to “5>=”.

7. HTT interruptions (line 66): the motif structure of the HTT gene is (CAG)_nCAACAGCCA(CCG)_n. The loss of interruption variant associated with the earlier onset of HD is CAA → CAG and CCA → CCG which leads to (CAG)_n(CCG)_n. Please correct this.

8. Line 250: Revise "extremely expansion" to "extreme expansion"

Reviewer #2 (Remarks to the Author):

The study by Ichikawa, Asano and Morishita aims to determine the structure of tandem repeats (TRs) in 270 control Samples from Japan. They use PacBio/SMRT HiFi long-read sequencing and develop a mathematical model to impute repeat motif composition at individual TR locus and use the corresponding information to assess the variability of TRs at the genome-wide level. Their main finding is that many TRs in the genome show a more complex structure than that reported in the reference genome. This catalogue could be used to determine possible association of TRs with some diseases.

Although the approach and the question are both interesting, the current manuscript is very difficult to read, and the use of some terms such as “mosaic” to describe the findings, which mainly report variable motif composition rather than true somatic mosaicism is very confusing. Furthermore, there is no comparison with other methods or existing dataset, which would have been very useful. Below is a list of points that need to be addressed before publication.

Major points:

1) The authors use the term “mosaic” to refer to tandem repeat structure that are complex (i.e. composed of several motifs instead of one motif and different from the motif present in the reference genome). This is very confusing as mosaicism in genetics usually refers to the existence of different genetic variants in two or more cell populations of an organism. What the authors describe here if I understand correctly corresponds to the normal genetic variability /variation at TR loci. It is possible or even likely that mosaicism (i.e. different motif length and structure at TR loci in a single individual) also exists but the method used (PacBio sequencing of DNA extracted from B cells with a mean coverage of 7.5X) is absolutely insufficient to address this question. The authors should make this point clear and make the difference between genetic variability observed between individuals and genetic variability existing within the same individual (postzygotic somatic mosaicism).

2) It is not clear to me if the DNA used for long-read sequencing was extracted to immortalized B-cells (i.e., lymphoblasts) and this should be clarified. The immortalization process using EBV is a clonal process that possibly leads to the selection of specific TR alleles and does not always represent the original genotype(s). The authors should therefore clarify this point and discuss possible bias associated with using immortalized cells if this is what they used. If possible, a comparison using DNA directly extracted from blood for a few individuals would allow ensuring that results are comparable for most studied loci.

3) The authors do not differentiate short tandem repeats (2-6 nucleotide repeats, microsatellites) from minisatellites (7-100 pb) in their work. However, previous studies (see for instance Legendre et al., *Genome Res* 2007; Mitra et al. *Nature*. 2021 Jan; 589(7841): 246–250 and Huang et al. www.biorxiv.org/content/10.1101/2022.05.12.491726v1). have shown that microsatellites and minisatellites have different mutability rates and constraints. Simple tandem repeats are mainly composed of one motif, that can vary from one individual to the other, while tandem repeats involving larger motifs show a substantial variability in the repeat sequence. The existence of different motifs at simple tandem repeat loci is usually rare and described as “interruptions”. The authors should therefore use these definitions and stratify the TRs they analyze according to their size.

4) A comparison of the results obtained in the study with previously developed approaches would be extremely valuable. For instance, would it be possible to compare tools (ExpansionHunter denovo, STRling, ExSTra...) existing for short read sequencing, which are also able to look at repeat motif/length composition, and compare these tools with results obtained with PacBio sequencing? The authors wrote that consider using nanopore sequencing but did not use it because of the high error rate but nanopore sequencing has been successfully used to assess the structure of repeat expansions (see for instance Mizuguchi et al. *Brain* 2021) and it would also be useful to compare both methods. It is possible that PacBio HiFi sequencing, which is based on sequencing of circularized molecules of 15-20 kb leads to biases for tandem repeats exceeding this size.

Reviewer #3 (Remarks to the Author):

The authors attempted to describe the landscape of mosaic tandem repeats (TRs) within the human genome. They used PacBio HiFi sequences of 270 Japanese individual as their analysis data. They reported diversity within the ~8,000 loci they genotyped.

One of the biggest appeals is a algorithm is delineate the complex repeat patterns with mosaic TRs, as this is lacking in existing TR detection software. Although the algorithm is described in the Method section, the results of the performance are actually reported in a separate, unpublished article. In light of the relative shallow sequencing depth (~8x) of the data, I do have reservation about the reliability of the complex TR patterns reported. And considering sequence diversity among these TR loci is apparently the major finding reported, how can one be confident that this is not due to sequencing error or artefact? For example, in Figure 1a, the first subgroup of group II only has a count of "one". Moreover, this particular locus, and some other loci of the 8,000 were found to reside in segmental duplications. This casts some doubts on the reliability of the read-anchoring part of the pipeline.

Speaking of the pipeline, it would be beneficial to the readers if the authors can put together a flow chart outlining the different analysis stages, and what software were applied in each stage.

Besides the methodology, some thoughts on a couple of analysis:

1. The idea of using surrounding SNVs to impute TR representatives is a bit unclear to me. Are the authors attempting to use neighbouring SNVs to group the TR representatives into haplotypes? And also what is the exact definition and significance of TR representatives and TR groups?
2. In the last section on disease association, for the "extreme expanded TRs" how much are they "expanded"? Are they expansions or just polymorphism within the human population?

We are grateful to the reviewers for their many valuable comments and suggestions on the manuscript. Accordingly, the manuscript, figures, and tables have been substantially revised. Revisions are highlighted in yellow in the new manuscript. In the following, we describe our responses to all comments and suggestions.

Reviewer #1

Comment: The authors of this study have analysed the PacBio HiFi data of 270 Japanese controls and have created a catalog of mosaic TRs. The findings of this study are very interesting and illuminate an important, yet unexplored feature of TRs. As the authors note, interrupting motifs in TR sequences have significant clinical relevance (e.g., the loss of CAA and CCA interruptions in the HTT gene can cause earlier onset of Huntington disease). Although software like ExpansionHunter (<https://github.com/Illumina/ExpansionHunter>) and REViewer (<https://github.com/Illumina/REViewer>) can characterize the motif composition of TRs and identify the presence of interrupting repeat motifs in the standard Illumina short-read sequencing data, respectively, the read length poses significant challenges in interrogating long TRs and minisatellites and their full-length motif structures genome-wide. The sequencing accuracy and read length of PacBio HiFi make it an ideal technology to study the length and motif composition of complex TR sequences. The authors have created a database of mosaic TRs (8,820 loci) that they highlight could be a valuable resource to study disease associations.

This is a very well-written manuscript with clearly presented data.

Answer: Thank you very much for valuable comments and suggestion. We have revised the manuscript accordingly in what follows.

Comment: 1. The database of 8,820 TR loci in Extended data table 1 could include additional annotations (gene names, location within the gene (intron, exon, UTR, etc.), any known disease associations, and/or overlaps with regulatory elements). This would help us better understand the diversity of TR alleles within different genomic contexts.

Answer: We are grateful to you for valuable suggestions. We have added gene annotation (exon, intro, UTR, etc) to TR loci in Extended Data Table 1 using the UCSC hg38 tables:

<https://hgdownload.soe.ucsc.edu/goldenPath/hg38/database/kgXref.txt.gz>

<https://hgdownload.soe.ucsc.edu/goldenPath/hg38/bigZips/genes/hg38.knownGene.gtf.gz>

We used geneSymbol in the first table above if it was present and spID in the second table otherwise.

Comment: 2. Although this study focuses on control samples, it would still be extremely valuable to characterize the allelic diversity and motif compositions of the 60 known disease-associated TR loci that may not harbor extreme expansions. Stevanovski et al. 2022 (PMID: 35245110 <https://pubmed.ncbi.nlm.nih.gov/35245110/>) reported diverse TR alleles with pentanucleotide motifs and identified several unique repeat motifs in their analysis of clinically nonaffected individuals. I suggest including the TR sizes and motif compositions of the 60 known disease loci of this cohort.

Answer: Thank you very much for this suggestion. We examined the following well maintained GnomAD table of 60 disease-associated TR regions that were generated from short read sequencing data by ExpansionHunter.

https://gnomad.broadinstitute.org/short-tandem-repeats?dataset=gnomad_r3

To demonstrate that this study augments the above information, the TR patterns observed in this study were added to each of the 60 entries, and the result is found in Extended Data Table 2. According to the known rules for testing whether a TR is pathogenic or not, some individuals had dominant alleles with expanded repeats in the *ATXN2*, *ATXN8OS*, *DABI*, and *SAMD12* gene coding regions, suggesting the individuals are carriers.

Stevanovski et al. examined a cohort of 37 individuals with 25 neurogenetic diseases and identified repeat expansions with 500 or more repeat units in *C9orf72*, *DABI*, *DMPK*, and *FXN*. These expansions were pathogenic and are not observed in our control samples (Extended Data Table 2). In *RFMI*, Stevanovski et al. reported (AAGGG)_{>500} repeat expansions, while we found (ACAGG)_{>500} repeat expansions, in which different units are expanded. These differences may be seen because Stevanovski et al. and we used case and control samples, respectively.

Comment: For normal TR alleles that can be spanned entirely by reads, ExpansionHunter calls from short reads are quite reliable. And gnomAD database includes the ExpansionHunter genotype calls as well as the read pileups of the disease TRs (https://gnomad.broadinstitute.org/short-tandem-repeats?dataset=gnomad_r3), and this could be a useful resource for comparing the observations in this

cohort vs other populations. And perhaps the authors could also mention the gnomAD resource for disease TR loci in their discussion.

Answer: ExpansionHunter and ExpansionHunter Denovo are able to identify representative units and their frequencies in short reads (<https://github.com/Illumina/ExpansionHunterDenovo>), but they are not designed to detect complex tandem repeat patterns with different units from long leads. For example, in *AFF2* gene, two patterns (GT)₁₂(CCG)₇₃ and (GT)₁₂(CCG)₅₅(AGCC)₅(CCG)₉ are frequently observed in our study, but only CGG-repeats are registered in gnomAD (see Extended Data Table 2), demonstrating that our study can complement complex tandem repeats missing in gnomAD.

Comment: 3. Have the authors used other TR-specific genotyping tools to characterize the expanded TRs in their dataset, or as a confirmation of their repeat size calls? For example, TRGT (<https://github.com/pacificBiosciences/trgt/>). Could the authors comment on the general strategies they had adopted to confirm or verify their genotype calls?

Answer: TRGT is a useful tool that is able to annotate tandem repeats in personal genomes accurately using a given tandem repeat pattern, and hence, it essentially requires a set of tandem repeat patterns as input. Thus, the set of tandem repeat patterns identified in this study should be a useful input to TRGT. We mentioned this research direction in the discussion of the revision.

Comment: 4. It would be really interesting to look at the methylation profiles of the expanded TRs the authors have identified in this study. Have the authors looked at the methylation characteristics of the expanded TRs? I suggest including this data.

Answer: Thank you very much for remarking this point. To detect CpG methylation from PacBio HiFi reads, we have used primrose (<https://github.com/PacificBiosciences/primrose>); however, no major changes in CpG methylation were observed in the extended TRs. The female samples showed X inactivation in genes *AFF2*, *AR*, *ARX1*, *FMRI*, and *SOX3* on the X chromosome, which are well known in the literature of studies on CpG methylation and are not highlighted in the revised version.

Comment: 5. The authors list the TR candidates in the hg38 reference genome. Could the authors remark on the TR loci and their motif compositions in the hg38 vs the T2T reference sequence?

Answer: TR loci in the hg38 reference genome were associated with their corresponding TR loci in the T2T reference genome, and ~1.97 million pairs of TR loci were found. Of these, 9,023 pairs differed by more than 1,000 nt in length and were further analyzed. TR loci in the hg38 genome were likely to be shorter than their corresponding loci in the T2T genome (Extended Data Figure 3f), though their repeat units were mostly quite similar. Like TR loci in the Japanese population, some TR loci in the T2T genome had motifs different from those in the hg38 genome.

Comment: 6. Fig 2c. Revise “5<=” to “5>=”.

Answer: Thank you for pointing this out. We have corrected the error to make it more understandable by replacing “5<=” with “5 or more.”

Comment: 7. HTT interruptions (line 66): the motif structure of the HTT gene is (CAG)_nCAACAGCCA(CCG)_n. The loss of interruption variant associated with the earlier onset of HD is CAA -> CAG and CCA -> CCG which leads to (CAG)_n(CCG)_n. Please correct this.

Answer: We are grateful to you for this remark, and have revised the sentence accordingly.

Comment: 8. Line 250: Revise "extremely expansion" to “extreme expansion”

Answer: We have revised the phrase accordingly.

Reviewer #2

Comment: The study by Ichikawa, Asano and Morishita aims to determine the structure of tandem repeats (TRs) in 270 control Samples from Japan. They use PacBio/SMRT HiFi long-read sequencing and develop a mathematical model to impute repeat motif composition at individual TR locus and use the corresponding information to assess the variability of TRs at the genome-wide level. Their main finding is that many TRs in the genome show a more complex structure than that reported in the reference genome. This catalogue could be used to determine possible association of TRs with some diseases. Although the approach and the question are both interesting, the current manuscript is very difficult to read, and the use of some terms such as “mosaic’ to describe the findings, which mainly report variable motif composition rather than true somatic mosaicism is very confusing.

Answer: We thank the reviewer for this valuable comment. Although the word “mosaic” is used in the computational biology community because it was used in:

Bankevich, A., Pevzner, P. (2020). MosaicFlye: Resolving Long Mosaic Repeats Using Long Reads. In: Schwartz, R. (eds) RECOMB 2020. Lecture Notes in Computer Science, vol 12074. Springer.

However, the word can be indeed confused with somatic mosaicism. Thus, we replaced all occurrences of “mosaic tandem repeats” with “complex tandem repeats.”

Comment: Furthermore, there is no comparison with other methods or existing dataset, which would have been very useful. Below is a list of points that need to be addressed before publication.

Answer: Although comparison with RepeatMasker were described in the previous version (Extended Data Figure 3bc), according to the comment, we added to the revision comparison with Tandem Repeat Finder (TRF) (Extended Data Figure 3de), and ExpansionHunter in gnomAD (Extended Data Table 2). Tandem Repeat GenoTyper (TRGT) is a useful tool that is able to annotate tandem repeats in personal genomes accurately using a given tandem repeat pattern, and hence, it essentially requires a set of tandem repeat patterns as input. Thus, the set of tandem repeat patterns identified in this study should be a useful input to TRGT. We mentioned this research direction in the discussion of the revision.

Comment: Major points: 1) the authors use the term “mosaic” to refer to tandem repeat structure that are complex (i.e. composed of several motifs instead of one motif and different from the motif present in

the reference genome). This is very confusing as mosaicism in genetics usually refers to the existence of different genetic variants in two or more cell populations of an organism. What the authors describe here if I understand correctly corresponds to the normal genetic variability /variation at TR loci. It is possible or even likely that mosaicism (i.e. different motif length and structure at TR loci in a single individual) also exists but the method used (PacBio sequencing of DNA extracted from B cells with a mean coverage of 7.5X) is absolutely insufficient to address this question. The authors should make this point clear and make the difference between genetic variability observed between individuals and genetic variability existing within the same individual (postzygotic somatic mosaicism).

Answer: We studied normal genetic variation at tandem repeat loci rather than variation in somatic mosaicism that requires unacceptable amount of long read sequencing data as you remark. Thus, to avoid confusion, we replaced all occurrences of “mosaic tandem repeats” with “complex tandem repeats.” To distinguish two different haplotypes with normal genetic variation at tandem repeat loci, we have proved that ~8-fold read coverage is statistically sufficient using the Lander-Waterman statistics (Extended Data Figure 2).

Comment: 2) It is not clear to me if the DNA used for long-read sequencing was extracted to immortalized B-cells (i.e., lymphoblasts) and this should be clarified. The immortalization process using EBV is a clonal process that possibly leads to the selection of specific TR alleles and does not always represent the original genotype(s). The authors should therefore clarify this point and discuss possible bias associated with using immortalized cells if this is what they used. If possible, a comparison using DNA directly extracted from blood for a few individuals would allow ensuring that results are comparable for most studied loci.

Answer: Thank you very much for valuable comments. Accordingly, we have revised the manuscript to remark that specific TR alleles may be selected during the immortalization process by EBV and do not necessarily represent the original allele, and the original blood sample of immortalized B-cells could not be obtained.

Comment: 3) The authors do not differentiate short tandem repeats (2-6 nucleotide repeats, microsatellites) from minisatellites (7-100 pb) in their work. However, previous studies (see for instance Legendre et al., *Genome Res* 2007; Mitra et al. *Nature*. 2021 Jan; 589(7841): 246–250 and Huang et al. www.biorxiv.org/content/10.1101/2022.05.12.491726v1). have shown that microsatellites and

minisatellites have different mutability rates and constraints. Simple tandem repeats are mainly composed of one motif, that can vary from one individual to the other, while tandem repeats involving larger motifs show a substantial variability in the repeat sequence. The existence of different motifs at simple tandem repeat loci is usually rare and described as “interruptions”.

Answer: The studies in the above publications used short-read sequencing data and were unable to precisely determine mini-satellites and complex tandem repeats (with interruptions) of length 100 bases or more. This caveat is found in Mitra et al. Nature. 2021 Jan; 589(7841):

“Our stringent filtering of input genotypes and resulting mutations is unlikely to capture large repeat expansions, which are often not supported by enclosing reads because the resulting alleles are longer than Illumina read lengths.”

Thus, although previous studies have primarily examined simple tandem repeats with a single motif less than 100 bases long, they have largely overlooked longer complex tandem repeats. Using long reads, our highly sensitive software program has shown that complex tandem repeats are much more abundant than simple tandem repeats (Figure 3c) and are more variable than minisatellites are (Figures 3ef).

Comment: The authors should therefore use these definitions and stratify the TRs they analyze according to their size.

Answer: Accordingly, we generated Figures 3c-f to show two groups of tandem repeats, short tandem repeats and minisatellites.

Comment: 4) A comparison of the results obtained in the study with previously developed approaches would be extremely valuable. For instance, would it be possible to compare tools (ExpansionHunter denovo, STRling, ExSTra...) existing for short read sequencing, which are also able to look at repeat motif/length composition, and compare these tools with results obtained with PacBio sequencing?

Answer: Thank you very much for this valuable suggestion. We examined the following well maintained GnomAD table of 60 disease-associated TR regions that were generated from short read sequencing data by ExpansionHunter.

https://gnomad.broadinstitute.org/short-tandem-repeats?dataset=gnomad_r3

To demonstrate that this study augments the above information, the TR patterns generated in this study were added to each of the 60 entries, and the result is found in Extended Data Table 2. According to the

known rules for testing whether a TR is pathogenic or not, some healthy Japanese individuals had dominant alleles with expanded repeats in the *ATXN2*, *ATXN8OS*, *DABI*, and *SAMD12* gene coding regions, suggesting the individuals are carriers.

Comment: The authors wrote that consider using nanopore sequencing but did not use it because of the high error rate but nanopore sequencing has been successfully used to assess the structure of repeat expansions (see for instance Mizuguchi et al. Brain 2021) and it would also be useful to compare both methods.

Answer: Mizuguchi et al. (Brain 144: 1103–1117, 2021) examined only the disease-associated (TTTCA/TTTA)-repeats found in the 4th intron of *SAMD12*. The pathogenic repeats are quite familiar to us since we first reported the repeats in our 2018 Nature Genetics paper together with Tsuji's group. Mizuguchi et al. enriched repeat regions using Cas9 target enrichment and obtained an extremely large number of Nanopore reads (~470x read coverage of the focal region), as Nanopore reads are quite error-prone (error rate of about 5%) and serious error correction to correctly infer repeat regions would require a huge amount of Nanopore reads. They also had to reconfirm their estimate of the (TTTCA/TTTA)-repeats using RP-PCR. Overall, the approach taken by Mizuguchi et al. could handle only a single gene region and needed a tremendous amount of Nanopore reads to revise highly erroneous reads.

By contrast, our approach used PacBio reads that were almost free of errors (error rate of ~0.1%) and we found that 10-fold read coverage was sufficient to infer long tandem repeats, allowing us to collect ~2 million complex tandem repeat expansions in a genome-wide manner.

Comment: It is possible that PacBio HiFi sequencing, which is based on sequencing of circularized molecules of 15-20 kb leads to biases for tandem repeats exceeding this size.

Answer: It is hard for PacBio HiFi reads of 10-20 kb in size to precisely to determine the composition of tandem repeats exceeding this size.

Reviewer #3

Comment: The authors attempted to describe the landscape of mosaic tandem repeats (TRs) within the human genome. They used PacBio HiFi sequences of 270 Japanese individual as their analysis data. They reported diversity within the ~8,000 loci they genotyped. One of the biggest appeals is a algorithm is delineate the complex repeat patterns with mosaic TRs, as this is lacking in existing TR detection software. Although the algorithm is described in the Method section, the results of the performance are actually reported in a separate, unpublished article.

Answer: Let us clarify why we have published the analysis and performance of the proposed algorithm in a separate article. This study aimed to understand the characteristics of complex TRs in populations and required the development of several software programs. One of the programs was an accurate algorithm for retrieving complex TRs from PacBio HiFi reads, which required solving computer science problems, including mathematical analysis of the computational complexity inherent in complex TR retrieval, designing an efficient algorithm, and applying the proposed algorithm to synthetic data to verify that it is more accurate than previous tools. Our solutions to these technical questions are somewhat outside the scope of *Nature Communications* and more appropriate for readers of *Bioinformatics*, and were reported in that journal in April 2023 (10.1093/bioinformatics/btad185). A PDF of this companion paper is attached to the revised manuscript, and its overview is outlined in the revised version.

Comment: In light of the relative shallow sequencing depth (~8x) of the data, I do have reservation about the reliability of the complex TR patterns reported. And considering sequence diversity among these TR loci is apparently the major finding reported, how can one be confident that this is not due to sequencing error or artefact? For example, in Figure 1a, the first subgroup of group II only has a count of "one".

Answer: Using the Lander-Waterman statistics, we have proved that ~8-fold read coverage is statistically sufficient to distinguish two different haplotypes with tandem repeats of ~5 kb in size (Extended Data Figure 2); however, the proof might be hard to find in the previous version and is therefore emphasized in the revision. Each TR member in the group is supported by multiple (typically four) high-precision PacBio HiFi leads with 99.9% accuracy, guaranteeing the TR's high accuracy.

Regarding the reliability of complex TR patterns, we have confirmed that our program uTR outperforms TRF and RepeatMasker using synthetic benchmark datasets of various types of complex TRs, which is

reported in the aforementioned *Bioinformatics* paper. Extended Data Figures 3d and 3e also explain the consistency of our program with the TRF for the detection of TRs in the reference human genome (hg38). We have emphasized these points in the revised version.

Comment: Moreover, this particular locus, and some other loci of the 8,000 were found to reside in segmental duplications. This casts some doubts on the reliability of the read-anchoring part of the pipeline.

Answer: This comment is truly valuable. Of 2,202,622 TR loci identified, 121556 (~5.5%) were located in segmental duplications in the hg38 reference (hg38_genomicSuperDups_2014-10-14), so to avoid false read alignments, segmental duplications were excluded from the analysis in this study. In Figure 1a, an example inside a segmental duplication is replaced by another example outside all segmental duplications.

Comment: Speaking of the pipeline, it would be beneficial to the readers if the authors can put together a flow chart outlining the different analysis stages, and what software were applied in each stage.

Answer: Thank you very much for this suggestion. We added Extended Data Figure 8 to show the pipeline accordingly.

Question: Besides the methodology, some thoughts on a couple of analysis: 1. The idea of using surrounding SNVs to impute TR representatives is a bit unclear to me. Are the authors attempting to use neighbouring SNVs to group the TR representatives into haplotypes?

Answer: We clustered the TR representatives that have identical pairs of closest SNVs into an identical group to highlight that TR representatives in the sample group can be very divergent (Figure 1d).

Question: And also what is the exact definition and significance of TR representatives and TR groups?

Answer: We first divided TR alleles collected from all individuals into clusters in terms of sequence dissimilarity (the edit distance between a pair of TRs to the centroid length) such that the dissimilarity between any pairs of members in each cluster is 3% or less. The centroid in each cluster is called the TR representative of the cluster. Afterward, we classified the TR representatives into TR groups that share the same pairs of nearest neighbor SNVs. The significance of TR groups is to highlight the difficulty of imputing a number of different TR representatives from surrounding identical SNVs.

Question : 2. In the last section on disease association, for the "extreme expanded TRs" how much are they "expanded"?

Answer: They are expanded by more than 100 bases from the median.

Question: Are they expansions or just polymorphism within the human population?

Answer: This is an important question. Some expansions are seen frequently and can be treated as polymorphism; however, most of extremely expanded tandem repeats are rare and should be treated as variants.

REVIEWERS' COMMENTS

Reviewer #1 (Remarks to the Author):

Thank you for addressing my comments and suggestions. I now only have a very minor suggestion and that is regarding the TRGT citation. The authors may consider citing the TRGT preprint (<https://www.biorxiv.org/content/10.1101/2023.05.12.540470v1>) in place of the GitHub link.

Reviewer #2 (Remarks to the Author):

The revision of the manuscript by Ichikawa and collaborators has clarified most of the point raised during the first review. The weak point of the study remains the low coverage of the HiFi data, that allows to obtain very few reads covering the expanded alleles and does not permit to address the somatic variability. Therefore, it is not clear if the sequences reported by the authors at these loci are the main repeat structures or only one of other possible structures existing at these loci in these individuals. However, the results are interesting as they clearly show that accurate long-read sequencing can accurately characterise variable tandem repeats and reveal complex alleles composed of different motifs that are typically not detected by short-read sequencing.

A couple of inaccuracies and typos should be corrected before publication:

- 1) Page 4, line 117: “biallelic AAAGG repeat expansion in associated with CANVAS” should be replaced by “biallelic AAGGG repeat expansion in associated with CANVAS” (AAGGG is the pathogenic motif, not AAAGG)
- 2) Page 14, line 281: RFM1 should be replaced by RFC1
- 3) Extended Data Table 1: column K correct Avgerage (Average); column L: correct Num ber (Number)
- 4) Extended Data Table 2: column G: correct Refernce (Reference). I suggest that the authors carefully read the manuscript and the figures/tables to correct possible additional typos
- 5) Extended Data Table 2: Pure TTTTA expanded alleles are not pathogenic at the SCA37 and SAMD12 loci. Only expansions containing TTTC A repeats are pathogenic. This should be corrected and clarified (no red color)
- 6) Pathogenic alleles in the AR gene: Are the corresponding individuals male or females?

7) The authors report 5 individuals with a SCA8 pathogenic expansion out of 270 individuals. Could the authors clarify if these individuals are affected and/or comment on the unexpected high frequency of this pathogenic expansion in control subjects?

Reviewer #3 (Remarks to the Author):

I want to thank the authors for doing the due diligence in response to my comments.

We thank the reviewers for valuable comments and suggestions on the manuscript. In the following, we describe our responses to all comments and suggestions.

Reviewer #1

Comment: Thank you for addressing my comments and suggestions. I now only have a very minor suggestion and that is regarding the TRGT citation. The authors may consider citing the TRGT preprint (<https://www.biorxiv.org/content/10.1101/2023.05.12.540470v1>) in place of the GitHub link.

Answer: We have cited the preprint accordingly.

Reviewer #2

Comment: The revision of the manuscript by Ichikawa and collaborators has clarified most of the point raised during the first review. The weak point of the study remains the low coverage of the HiFi data, that allows to obtain very few reads covering the expanded alleles and does not permit to address the somatic variability. Therefore, it is not clear if the sequences reported by the authors at these loci are the main repeat structures or only one of other possible structures existing at these loci in these individuals.

Answer: We agreed with these points and accordingly further detailed some caveats and limitations regarding the study design in the discussion (highlighted in yellow in the revised manuscript):

“Generation of fully haplotype-resolved diploid human genome assemblies is resource-intensive, requiring 30–40-fold coverage of HiFi reads from single individuals and high coverage of Hi-C and Nanopore sequencing data. We may need >1000-fold coverage of HiFi reads to address somatic variability in TRs³⁴. Therefore, there is uncertainty as to whether the sequence reported in this study is the predominant repetitive structure or just one of other structures that may be present at these loci in these individuals. In this study, to acquire data from a larger number of individuals, one single-molecule real-time sequencing cell (SMRT cell) was used to obtain moderate coverage (≥ 7.5) reads, which, in theory, allowed reliable observation of the majority of complex TRs (Methods, Supplementary Fig. 2).”

Comment: However, the results are interesting as they clearly show that accurate long-read sequencing can accurately characterise variable tandem repeats and reveal complex alleles composed of different motifs that are typically not detected by short-read sequencing.

Answer: Thank you for the comments on our manuscript.

Comment: A couple of inaccuracies and typos should be corrected before publication:

Answer: We are grateful to the reviewer for carefully reading our manuscript and pointing out these mistakes. We revised Supplementary Data 1 and 2, and re-annotated the table in Supplementary Data 2 for better understanding.

Comment:

1) Page 4, line 117: “biallelic AAAGG repeat expansion in associated with CANVAS” should be replaced by “biallelic AAGGG repeat expansion in associated with CANVAS” (AAGGG is the pathogenic motif, not AAAGG).

2) Page 14, line 281: RFM1 should be replaced by RFC1

3) Extended Data Table 1: column K correct Avgerage (Average); column L: correct Num ber (Number)

4) Extended Data Table 2: column G: correct Refernce (Reference). I suggest that the authors carefully read the manuscript and the figures/tables to correct possible additional typos

Answer: We have revised these errors accordingly. A few other minor typos have also been fixed.

Comment: 5) Extended Data Table 2: Pure TTTTA expanded alleles are not pathogenic at the SCA37 and SAMD12 loci. Only expansions containing TTTC A repeats are pathogenic. This should be corrected and clarified (no red color)

Answer: Thank you very much for pointing out these mistakes. We have amended them in the main text and listed these points in Column F of the table in Supplementary Data 2.

Comment: 6) Pathogenic alleles in the AR gene: Are the corresponding individuals male or females?

Answer: All samples are male. To collect information on the Y chromosome, 258 of the 270 control samples are male, which is mentioned in the revised manuscript. We also precisely re-analyzed expanded AGC repeats that appeared pathogenic or abnormal to find that some of them had interruptions and were not pathogenic. This issue is Occam’s razor: tradeoff between simpler patterns with more mismatches and more complex patterns with less mismatches. To point out this problem, we re-examined all expanded repeats in the table in Supplementary Data 2 and annotated some repeats with simple and complex patterns using our program. Patterns requiring attention are highlighted in orange in the H column of the table.

Comment: 7) The authors report 5 individuals with a SCA8 pathogenic expansion out of 270 individuals. Could the authors clarify if these individuals are affected and/or comment on the unexpected high frequency of this pathogenic expansion in control subjects?

Answer: Information about whether individuals have a particular disease is anonymized. We do not have a clear answer to the unexpectedly high frequency. Koob et al. (*Nature Genetics* 21, 379–384, 1999) mentioned “All but one of the individuals with a CTG repeat tract of more than 107 repeats are clinically affected. This 42-year-old individual has 140 CTG repeats,” suggesting that some healthy individuals can have a CTG repeat tract of more than 107 repeats.

Reviewer #3 (Remarks to the Author):

Comment: I want to thank the authors for doing the due diligence in response to my comments.

Answer: Thank you for your valuable comments that helped improve the manuscript.